# Scanned optogenetic control of mammalian somatosensory input to map input-specific behavioral outputs

Ara Schorscher-Petcu, Flóra Takács, Liam E Browne*

Wolfson Institute for Biomedical Research, and Department of Neuroscience, Physiology and Pharmacology, University College London, London, United Kingdom

**Abstract** Somatosensory stimuli guide and shape behavior, from immediate protective reflexes to longer-term learning and higher-order processes related to pain and touch. However, somatosensory inputs are challenging to control in awake mammals due to the diversity and nature of contact stimuli. Application of cutaneous stimuli is currently limited to relatively imprecise methods as well as subjective behavioral measures. The strategy we present here overcomes these difficulties, achieving 'remote touch' with spatiotemporally precise and dynamic optogenetic stimulation by projecting light to a small defined area of skin. We mapped behavioral responses in freely behaving mice with specific nociceptor and low-threshold mechanoreceptor inputs. In nociceptors, sparse recruitment of single-action potentials shapes rapid protective pain-related behaviors, including coordinated head orientation and body repositioning that depend on the initial body pose. In contrast, activation of low-threshold mechanoreceptors elicited slow-onset behaviors and more subtle whole-body behaviors. The strategy can be used to define specific behavioral repertoires, examine the timing and nature of reflexes, and dissect sensory, motor, cognitive, and motivational processes guiding behavior.

**\*For correspondence:**
liam.browne@ucl.ac.uk

**Competing interest:** The authors declare that no competing interests exist.

## Introduction

The survival of an organism depends on its ability to detect and respond appropriately to its environment. Afferent neurons innervating the skin provide sensory information to guide and refine behavior (*Seymour, 2019*; *Zimmerman et al., 2014*). Cutaneous stimuli are used to study a wide range of neurobiological mechanisms since neurons densely innervating skin function to provide diverse information as the body interfaces with its immediate environment. These afferents maintain the integrity of the body by recruiting rapid sensorimotor responses, optimize movement through feedback loops, provide teaching signals that drive learning, and update internal models of the environment through higher-order perceptual and cognitive processes (*Barik et al., 2018*; *Brecht, 2017*; *Corder et al., 2019*; *de Haan and Dijkerman, 2020*; *Haggard et al., 2013*; *Huang et al., 2019*; *Petersen, 2019*; *Seymour, 2019*). Damaging stimuli, for example, evoke rapid motor responses to minimize immediate harm and generate pain that motivates longer-term behavioral changes.

Compared to visual, olfactory, and auditory stimuli, somatosensory inputs are challenging to deliver in awake unrestrained mammals. This is due to the nature of stimuli that require contact and the diversity of stimulus features encoded by afferents that innervate skin. Cutaneous afferent neurons are functionally and genetically heterogeneous, displaying differential tuning, spike thresholds, adaptation rates, and conduction velocities (*Abraira and Ginty, 2013*; *Dubin and Patapoutian, 2010*; *Gatto et al., 2019*; *Häring et al., 2018*). The arborization of their peripheral terminals can delineate spatial and temporal dimensions of the stimulus (*Pruszynski and Johansson, 2014*), particularly once many inputs are integrated by the central nervous system (*Prescott et al., 2014*). Cutaneous stimulation in freely moving mice often requires the experimenter to manually touch or approach

**eLife digest** To safely navigate their world, animals need to be able to tell apart a gentle touch from an eye-watering pinch, detect cold water or sense the throbbing pain stemming from an infected cut. These 'somatic' sensations are relayed through thousands of nerve endings embedded in the skin and other tissues. Yet the neurological mechanisms that underpin these abilities are complex and still poorly understood.

Indeed, these nerve endings can be stimulated by extreme temperatures, harmful chemicals, friction or even internal signals such as inflammation. One event can also recruit many different types of endings: a cut for example, will involve responses to mechanical pressure, tissue damage and local immune response. To disentangle these different actors and how they affect behavior, scientists need to develop approaches that allow them to deliver specific stimuli with increased precision, and to monitor the impact on an animal.

To achieve this goal, Schorscher-Petcu et al. used mice in which blue light could trigger specific types of nerve endings. For instance, depending on the genetic background of the animals, a laser could either activate nerve endings involved in pain or gentle touch. Crucially, this could be done from a distance by beaming light with exquisite precision onto the paws of the mice without physically touching or disturbing the animals.

How the mice responded could then be observed without any interference. Their behavior was analyzed using a combination of high-speed videos, computer-driven recording systems, and machine learning. This revealed subtle changes in behavior that had not been detected before, spotting microscopic movements of the stimulated paw and mapping simultaneous whole-body movements such as changes in posture or head orientation. The approach therefore allows scientists to assess the impact of touch, pain or temperature sensation in freely behaving mice. It could also be harnessed to develop much needed treatments against chronic pain.

the skin. This results in inaccurate timing, duration, and localization of stimuli. The close proximity of the experimenter can cause observer-induced changes in animal behavior (*Sorge et al., 2014*). Stimuli also activate a mixture of sensory neuron populations. For example, intense stimuli can co-activate fast-conducting low-threshold afferents that encode innocuous stimuli simultaneously with more slowly conducting high-threshold afferents (*Wang et al., 2018*). The latter are nociceptors that trigger fast protective behaviors and pain. Consequently, mixed cutaneous inputs recruit cells, circuits, and behaviors that are not specific to the neural mechanism under study. A way to control genetically defined afferent populations is to introduce opsins into these afferents and optogenetically stimulate them through the skin (*Abdo et al., 2019*; *Arcourt et al., 2017*; *Barik et al., 2018*; *Beaudry et al., 2017*; *Browne et al., 2017*; *Daou et al., 2013*; *Iyer et al., 2014*). However, these methods in their current form do not fully exploit the properties of light.

The behaviors that are evoked by cutaneous stimuli are also typically measured with limited and often subjective means. Manual scoring introduces unnecessary experimenter bias and omits key features of behavior. Behavioral assays have traditionally focused on a snapshot of the stimulated body part rather than dynamics of behavior involving the body as a whole (*Gatto et al., 2019*). Recent advances in machine vision and markerless pose estimation have enabled the dissection of animal behavioral sequences (*Mathis et al., 2018*; *Pereira et al., 2019*; *Wiltschko et al., 2015*). However, these have not been adapted to study behavioral outputs relating to specific cutaneous inputs.

Here we developed an approach to project precise optogenetic stimuli onto the skin of freely behaving mice (*Figure 1A*). The strategy elicits time-locked individual action potentials in genetically targeted afferents innervating a small stimulation field targeted to the skin. Stimuli can be delivered remotely as predefined microscale patterns, lines, or moving points. The utility of the system was demonstrated by precisely stimulating nociceptors, or Aβ low threshold mechanoreceptors (LTMRs), in freely behaving mice to map behavioral outputs at high speed. We provide an analysis toolkit that quantifies the millisecond-timescale dynamics of behavioral responses using machine vision methods. We dissect discrete behavioral components of local paw responses, head orienting and body repositioning behaviors, and determine how these specific behavioral components relate to precise somatosensory inputs.

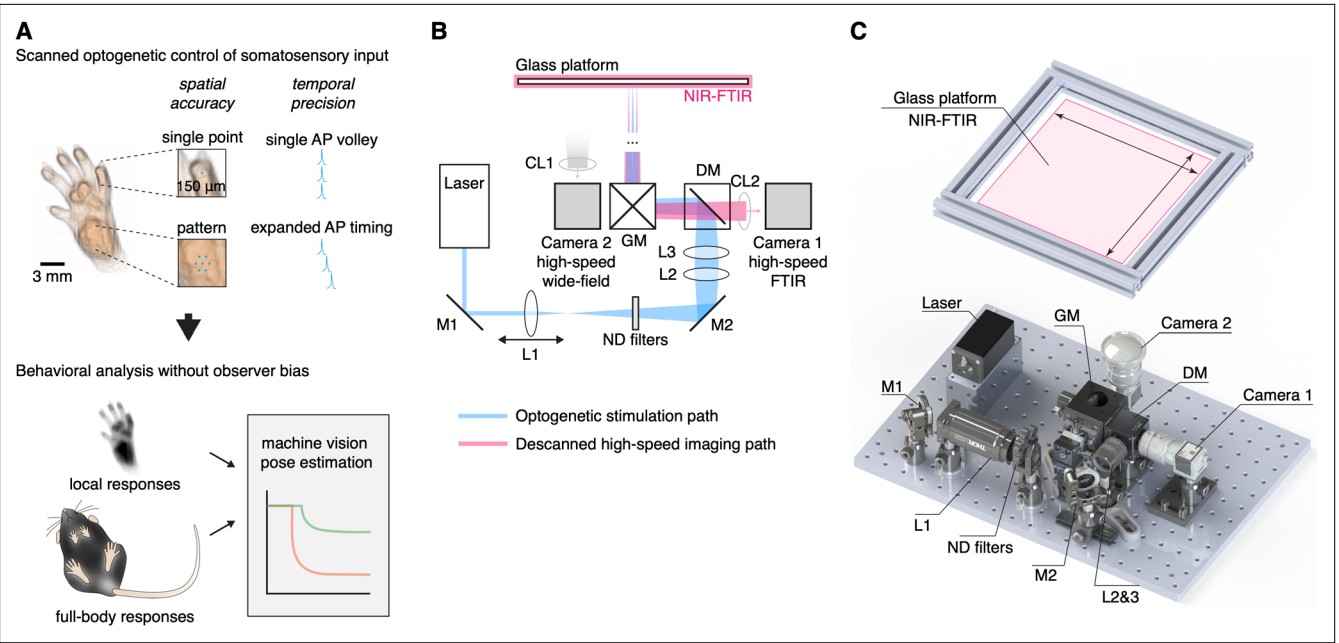

**Figure 1.** Remote and precise somatosensory input and analysis of behavior. (**A**) Afferent neurons expressing ChR2 are controlled remotely in freely behaving mice by projecting laser light with sub-millimeter precision to the skin. This enables precise non-contact stimulation with microscale patterns, lines, and points using scanned transdermal optogenetics. Time-locked triggering of single-action potential volleys is achieved through high temporal control of the laser. Behavioral responses can be automatically recorded and analyzed using a combination of computational methods. (**B**) Schematic of the stimulation laser (in blue) and infrared imaging (in red) paths. Mirrors (M1 and M2) direct the laser beam through a set of lenses (L1–L3), which allow the beam to be focused manually to pre-calibrated spot sizes. A dichroic mirror (DM) guides the laser beam into a pair of galvanometer mirrors, which are remotely controlled to enable precise targeting of the beam onto the glass platform. Near-infrared frustrated total internal reflection (NIR-FTIR) signal from the glass platform is descanned through the galvanometers and imaged using a high-speed infrared camera. A second wide-field camera is used to concomitantly record a below view of the entire glass platform. (**C**) Rendering of the assembled components. A Solidworks assembly is available at https://github.com/browne-lab/throwinglight.

The online version of this article includes the following figure supplement(s) for figure 1:

**Figure supplement 1.** Technical calibration of the optical system.

**Figure supplement 2.** Hardware and software information flow used in the optical system.

## Results

### Design of the optical approach

The design of the optical strategy had eight criteria: (1) that somatosensory stimuli are delivered non-invasively without touching or approaching the mice; (2) localization of stimuli are spatially precise and accurate (<10 μm); (3) freely moving mice can be targeted anywhere within a relatively large (400 cm$^2$) arena; (4) stimuli can be controlled with a computer interface from outside the behavior room; (5) stimulation patterns, lines, and points are generated by rapidly scanning the stimuli between predefined locations; (6) stimulation size can be controlled down to ≥150 μm diameter; (7) stimuli are temporally precise to control individual action potentials using sub-millisecond time-locked pulses; and (8) behavioral responses are recorded at high speed at the stimulated site and across the whole body simultaneously. An optical system was assembled to meet these specific criteria (*Figure 1B and C*).

The stimulation path uses two mirror galvanometers to remotely target the laser stimulation to any location on a large glass stimulation floor. A series of lenses expands the beam and then focuses it down to 0.018 mm$^2$ (150 μm beam diameter) at the surface of this floor. This was defocused to provide a range of calibrated stimulation spot sizes up to 2.307 mm$^2$, with separable increments that were stable over long periods of time (*Figure 1—figure supplement 1A*). The optical power density could be kept equal between these different stimulation spot sizes. The glass floor was far (400 mm) from the galvanometers, resulting in a maximum focal length variability of <1.5% (see Materials and methods). This design yielded a spatial targeting resolution of 6.2 μm while minimizing variability in laser stimulation spot sizes across the large stimulation plane (coefficient of variation ≤0.1,

*Figure 1—figure supplement 1B*). The beam ellipticity was 74.3% ± 14.3% (median± MAD, range of 36–99%) for all spot sizes. The optical power was uniform across the stimulation plane (*Figure 1—figure supplement 1C*). The galvanometers allow rapid small angle step (300 µs) responses to scan the laser beam between adjacent positions and shape stimulation patterns using brief laser pulses (diode laser rise and fall time: 2.5 ns). Custom software (see Materials and methods) was developed to remotely control the laser stimulation position, trigger laser pulses, synchronize galvanometer jumps, and trigger the camera acquisition (*Figure 1—figure supplement 2*).

The camera acquisition path was used to manually target the location of the laser stimulation pulse(s); the path was descanned through the galvanometers so that the alignment between the laser and camera is fixed (*Figure 1B*). The camera feed is displayed in the user interface and enables the operator to use this image to target the laser to the desired location. High signal-to-noise recordings were obtained using near-infrared frustrated total internal reflection (NIR-FTIR) in the glass stimulation floor (Roberson, D. P. et al., manuscript submitted). If a medium (skin, hair, tail, etc.) is within a few hundred microns of the glass, it causes reflection of the evanescent wave and this signal decreases non-linearly with distance from the glass such that very minor movements of the paw can be detected. The acquisition camera acquired the NIR-FTIR signal in high-speed (up to 1000 frames/s) with a pixel size of 110 µm. A second camera was used to record the entire arena and capture behaviors involving the whole body before and after stimulation. Offline quantification was carried out using custom analysis code combined with markerless tracking tools (*Mathis et al., 2018*).

## Mapping high-speed local responses to nociceptive input

To validate the strategy, we first crossed *Trpv1*-IRES-Cre (TRPV1$^{Cre}$) and R26-CAG-LSL-ChR2-tdTomato mice to obtain a line (TRPV1$^{Cre}$::ChR2) in which ChR2 is selectively expressed in a broad class of nociceptors innervating glabrous skin (*Browne et al., 2017*). These mice were allowed to freely explore individual chambers placed on the stimulation plane. When mice were idle (still and awake), a time-locked laser pulse was targeted to the hind paw. Stimuli could be controlled remotely from outside the behavior room. We recorded paw withdrawal dynamics with millisecond resolution. For example, a single, small 1 ms laser pulse initiated a behavioral response at 29 ms, progressing to complete removal of the hind paw from the glass floor just 5 ms later (*Figure 2A* and *Figure 2—video 1*). The stimulus used for this protocol was $S_6$, 0.577 mm$^2$ in area, which corresponds to less than 1 % of the glabrous paw area and highlights the sensitivity of the nociceptive system. Motion energy, individual pixel latencies, and response dynamics could be extracted from these high-speed recordings (*Figure 2B and C*).

We probed multiple sites across the plantar surface and digits and found that the hind paw heel gave the most robust responses (*Figure 2—figure supplement 1*). This region was targeted in all subsequent experiments. Littermates that did not express the Cre recombinase allele confirmed that the laser stimulation did not produce non-specific responses. These mice did not show any behavioral responses, even with the largest stimuli (spot size $S_8$, 30 ms pulse, *Figure 2—figure supplement 2*). We next provide some examples of the utility of the strategy by examining the relationship between nociceptive input and protective behaviors.

## Probabilistic nociceptor recruitment determines the nature, timing, and extent of behavior

Fast protective withdrawal behaviors can be triggered by the first action potential arriving at the spinal cord from cutaneous nociceptors. A brief optogenetic stimulus generates just a single-action potential in each nociceptor activated (*Browne et al., 2017*). This is due to the rapid closing rate of ChR2 relative to the longer minimal interspike interval of nociceptors. The same transient optogenetic stimulus (*Browne et al., 2017*), or a pinprick stimulus (*Arcourt et al., 2017*), initiates behavior before a second action potential would have time to arrive at the spinal cord. That the first action potential can drive protective behaviors places constraints on how stimulus intensity can be encoded, suggesting that the total population of nociceptors firing a single-action potential can provide information as a "Boolean array." The consequences of this have not been investigated previously as precise control of specific nociceptive input had not been possible. We predicted that the relative number of nociceptors firing a single-action potential determines the features of the behavioral response.

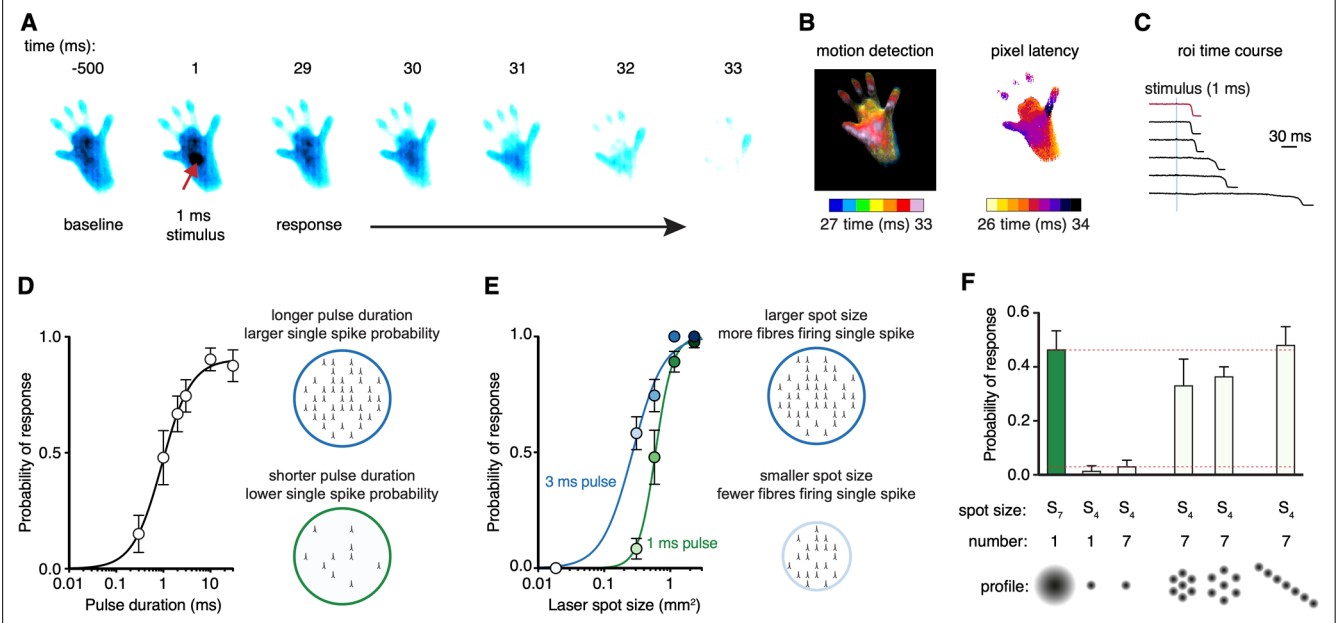

**Figure 2.** Scanned optogenetic stimuli reveal relationships with local behaviors. (**A**) Millisecond-timescale changes in hind paw near-infrared frustrated total internal reflection (NIR-FTIR) signal in response to a single 1 ms laser pulse (laser spot size $S_6$ = 0.577 mm²) recorded at 1000 frames/s. (**B**) Motion energy analysis (left) and response latencies calculated for each pixel (right) for the same trial as in (**A**). (**C**) Example traces of the NIR-FTIR signal time course as measured within a circular region of interest centered on the stimulation site. Six traces from two animals are depicted (1 ms pulse, spot size $S_6$ = 0.577 mm²). The red trace corresponds to the example trial illustrated in (**A**) and (**B**). (**D**) Paw response probability increases as a function of laser pulse duration when stimulation size is constant (spot size $S_6$ = 0.577 mm²; 37–42 trials for each pulse duration from eight mice, mean probability ± SEM). Light pulses 10 ms or less with the same intensity and wavelength have been shown to generate just a single-action potential in each nociceptor activated in the TRPV1^Cre::ChR2 line (*Browne et al., 2017*). Note that a 30 ms might generate more than one action potential but the response already plateaus at 10 ms duration, suggesting one action potential per nociceptor shapes the response. (**E**) Paw response probability increases as a function of laser stimulation spot size when pulse duration is constant. Data are 34–45 trials for each spot size per pulse duration from 7 to 8 mice, shown as mean probability ± SEM. The dataset for (**D**) and (**E**) is provided in *Figure 2—source data 1*. (**F**) Stimulation patterning shows that the absolute size, rather than the geometric shape, of the nociceptive stimulus determines the withdrawal probability (Friedman's non-parametric test for within subject repeated measures S(5) = 22.35, p=0.0004). Paw response probabilities in response to a single large laser spot ($S_7$ = 1.15 mm²), a single small spot ($S_4$ = 0.176 mm²; p=0.018 compared to $S_7$ and p=0.013 compared to the line pattern), a 10 ms train of seven small 1 ms spots targeting the same site (p=0.039, compared to $S_7$ and p=0.030 compared to the line pattern) or spatially translated to produce different patterns. Note that the cumulative area of the seven small spots approximates the area of the large spot, and no statistically significant difference was detected between any of their response probabilities. Data shown as mean probability ± SEM are from n = 6 mice, with each 6–10 trials per pattern. The dataset for (**F**) is provided in *Figure 2—source data 2*.

The online version of this article includes the following video and figure supplement(s) for figure 2:

**Source data 1.** Time courses of paw movement recorded at 1000 frames/s with stimuli that vary in duration and size.

**Source data 2.** Time courses of paw movement recorded at 500 frames/s with single point and patterned stimuli.

**Figure supplement 1.** Microscale mapping of sensitivity to noxious optogenetic stimulation.

**Figure supplement 2.** Littermate controls do not respond to optogenetic stimulation.

**Figure 2—video 1.** Pain-related hind paw withdrawals.

https://elifesciences.org/articles/62026/figures#fig2video1

Varying the pulse duration with nanosecond precision influences the probability of each nociceptor generating a single-action potential within the stimulation site. A pulse as short as 300 µs elicited behavioral responses but with relatively low probability (*Figure 2D*). This probability increased with pulse duration until it approached unity, closely matching the on-kinetics of the ChR2 used (τ = 1.9 ms; *Lin, 2011*). We next controlled the spatial, rather than temporal, properties of the stimulation in two further experiments. Firstly, we find that the total area of stimulated skin determines the behavioral response probability, such that the larger the nociceptive input the larger the response probability (*Figure 2E*). Secondly, we generated different stimulation patterns. We find that sub-threshold stimulations are additive (*Figure 2F*). Specifically, seven spatially displaced small sub-threshold stimulations could reproduce the response probability of a single large stimulation that was approximately seven

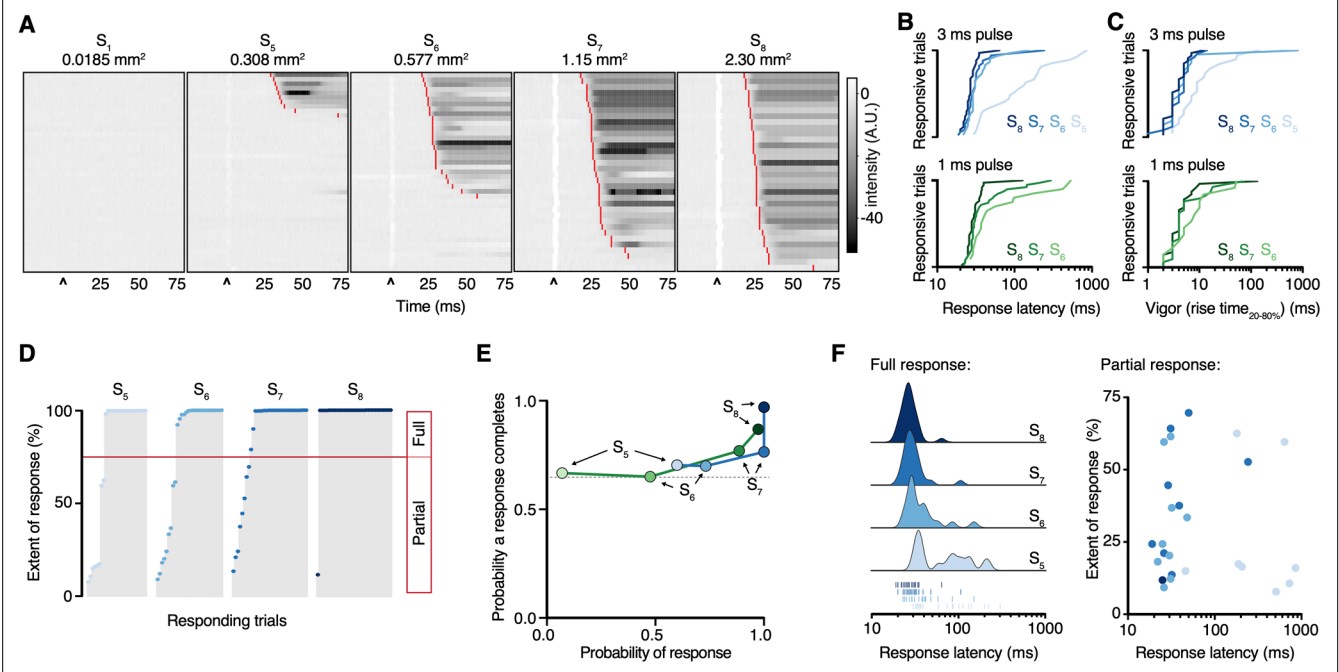

**Figure 3.** Paw response latency and magnitude are influenced by the sparse recruitment of nociceptors. (**A**) Raster plots of hind paw responses for five different 3 ms laser stimulation sizes, sorted by response latency. The paw response latency is indicated in red. (**B**) Paw response latencies to trials with single 3 ms (blue, top) and 1 ms (green, bottom) stimulations at different spot sizes, sorted by latency. (**C**) Response vigor (hind paw rise time, 20–80%) to single 3 ms (blue, top) or 1 ms (green, bottom) pulses with a range of stimulation spot sizes. Rise times to a 3 ms pulse were 4 ± 1 ms, 4 ± 1 ms, 4 ± 1 ms, and 9 ± 5 ms for spot sizes $S_8$, $S_7$, $S_6$, and $S_5$, respectively, and to a 1 ms pulse were 4 ± 1 ms, 5 ± 2 ms, and 6 ± 3 ms for spot sizes $S_8$, $S_7$, and $S_6$, respectively. (**D**) Extent of responses (%NIR-FTIR signal decrease). The threshold for a full response and partial response is 75 % of baseline signal (red line). (**E**) The probability of responses to reach completion (full response) as a function of the probability of response for four stimulation spot sizes and two pulse durations (green 1 ms; blue 3 ms). (**F**) Response latency distributions for trials that reach completion (full response) shown with Gaussian kernel density estimation of data (left). Rug plot inset representing individual response latencies for each color-coded spot size. No correlation was observed between response latency and extent for partial responses when stimulation duration was 3 ms. Data from 7 to 8 mice with 39–44 trials per spot size for 1 ms pulse duration and 34–44 per spot size for 3 ms pulse duration. The dataset is provided in *Figure 2—source data 1*. NIR-FTIR: near-infrared frustrated total internal reflection.

times their size. This could not be achieved by repeated application of the small stimulations to the same site (*Figure 2F*).

Time-locking the stimulus enabled us to examine the hind paw responses with high temporal resolution. The nociceptive input size influenced the behavioral response latency: for example, a 3 ms pulse resulted in response latencies of 27 ± 1 ms, 30 ± 2 ms, 33 ± 5 ms, and 112 ± 46 ms for spot sizes $S_8$, $S_7$, $S_6$, and $S_5$, respectively (*Figure 3A and B*). The shorter latencies are consistent with medium-conduction velocity Aδ-fibers that arrive at the spinal cord before slower C-fiber action potentials (>35 ms) (*Browne et al., 2017*). The rank order of response latencies follows the nociceptive input size for both pulse durations, and they fit well with log-log regressions (3 ms pulse $R^2$ = 0.87, 1 ms pulse $R^2$ = 0.90). Once a hind limb motor response was initiated, it developed rapidly, lifting from the glass with rise times that show the vigor of the motor response was also dependent on nociceptive input size (*Figure 3C*). These responses, in >65% of cases, proceeded to full withdrawal. However, in a fraction of trials the paw moved but did not withdraw (*Figure 3D*), highlighting the sensitivity of the acquisition system. Even the smallest of nociceptive inputs still produced a large fraction of full withdrawal responses, despite decreases in response probability (*Figure 3E*). The fraction of full withdrawal responses increased with the size of nociceptive input. The onset latency of both full and partial responses decreased as nociceptive input increased (*Figure 3F*).

## Whole-body behavioral responses to remote and precise nociceptive input

Pain-related responses are not limited to the affected limb but involve simultaneous movement of other parts of the body (*Blivis et al., 2017*; *Browne et al., 2017*). These non-local behaviors theoretically serve several protective purposes: to investigate and identify the potential source of danger, move the entire body away from this danger, attend to the affected area of the body (*Huang et al., 2019*) and to maintain balance (*Sherrington, 1910*). Whole-body movements were quantified as motion energy (*Figure 4—figure supplement 1A*) and high-speed recordings show this initiated with a mean response latency of 30 ± 1 ms, with the first movement bout displaying a mean duration of 136 ± 14 ms (80 trials from 10 mice) (*Figure 4—figure supplement 2*). The magnitude of whole-body movement increased with the stimulation spot size (*Figure 4—figure supplement 1B*). Peak motion energy had a lognormal relationship with nociceptive input size ($R^2$ = 0.99). This indicates that global behaviors are also proportional to the relative size of the nociceptive input; the recruited nociceptors firing a single-action potential (*Figure 4—figure supplement 1B*).

## Sparse nociceptor stimulation triggers coordinated postural adjustments

Most behaviors arise from the complex coordination of discrete body parts, which can be tracked individually. To dissect specific components of these behaviors, we implemented DeepLabCut (*Mathis et al., 2018*) by training a network using frames from the high-speed (400 frames/s) videos to track 18 user-defined body parts across the mouse (for details, refer to Materials and methods, *Global behaviors during optogenetic stimulation*). The high-speed video recordings of stimulation trials were analyzed using this network. Specific nociceptive input at the hind paw ($S_8$, 2.307 mm$^2$, 10 ms pulse) causes behavior that initiates simultaneously across the body. Inspection of the movements of each body part relative to the baseline pose (*Figure 4A*) shows fast outward movement of the stimulated and contralateral hind paws, and concomitant initiation of head orientation (two example responses in *Figure 4B*). Based on these observations, we examined the behavioral trajectories in the first 115 ms across the population of 80 trials. The first three principal components (PCs) were fit using six body part x and y values at 115 ms after the stimulus onset. These PCs explain 88.8 % of the variance (50.4, 26.5, and 11.9% for PC1, PC2, and PC3, respectively). PC1 is dominated by hind paw translation, PC2 by head and body movement, and PC3 by head orientation (*Figure 4C*). Projecting the entire time course onto these same PCs can explain 78.1 % of the variance (37.1, 24.3, and 16.7% for PC1, PC2, and PC3, respectively). The response trajectories revealed that movements occur largely in same direction within PC space with a circular standard deviation of 52.9° (*Figure 4D and E*). Shuffling body parts on each trial gave non-directional trajectories with a circular standard deviation of 126.8° (*Figure 4—figure supplement 3*). Behavioral trajectories also show that the response magnitude in PC space can be partly explained by initial PC1 and PC2 values (*Figure 4F and G*). This suggests that the initial pose influences these fast behavioral responses.

Examining specific features of these behaviors over a slightly longer period (300 ms) provides further insights. Displacement of each body part relative to their baseline position reveals the response timing, extent, and coordination (*Figure 4H*). The stimulated paw started moving at 29 ± 1 ms, the contralateral hind paw at 34 ± 4 ms, and the nose at 33 ± 2 ms (80 trials from 10 mice). With this intense stimulus, only in 6 % of trials did the hind paws or single body parts move alone, although the magnitude of the head movement varied between trials. The distance traveled by the nose positively correlates with the distance for the stimulated paw (Pearson's *r* = 0.64, n = 80 trials from 10 mice). Examining the relative distance between the nose and stimulated hind paw shows a reliably short latency (*Figure 4I*), indicating that these responses are driven by Aδ-nociceptor input rather than more slowly conducting C-fibers. A diversity of responses was observed: the head and stimulated paw move closer together in some trials and in others moved further apart (*Figure 4I and J*). This could result from the head moving towards or away from the stimulated paw but also the stimulated paw moving backwards as the body rotates. Indeed, consistent with initial observations (*Figure 4A and B*) and principal component analysis (PCA; *Figure 4C–G*), we find that the head selectively and rapidly orients to the stimulated side (*Figure 4K*). The presence of head orientation suggests that a brief nociceptive input can rapidly generate a coordinated spatially organized behavioral response. This is likely integral to protective pain-related behaviors and might function to gather sensory information

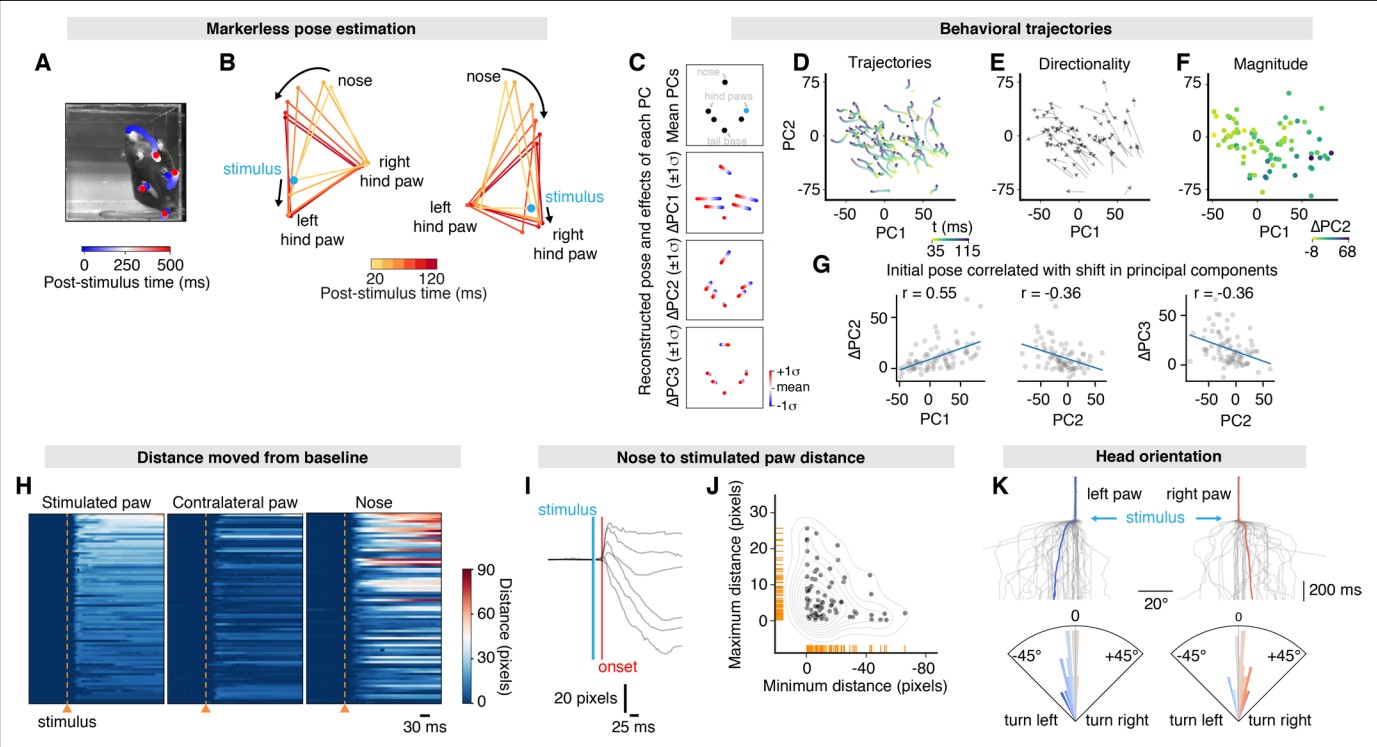

**Figure 4.** Mapping whole-body behavioral repertoires to precise nociceptive input. (**A**) Example spatiotemporal structure of a noxious stimulus response superimposed on the baseline image taken immediately before stimulus. The color indicates the timing of nose and hind paw trajectories. In this example, the left hind paw of the mouse was stimulated, which is the right hind paw as viewed in the image. For ease, we refer to the stimulation side as viewed in the image, rather than the side with respect to the mouse. (**B**) Example graphical representation showing the sequence of postural adjustment following nociceptive stimulus in two trials. Left: the left (as viewed) hind paw was stimulated. Right: the right (as viewed) hind paw was stimulated. (**C**) Principal component analysis of the x and y values for six body parts – nose, left hind paw digits, left hind paw heel, right hind paw digits, right hind paw heel, and tail base – across all 80 trials. Coordinates were egocentrically aligned by the baseline pose, setting the tail base as origin and the stimulated paw on the right. This allowed the reconstruction of these locations using the first three principal components (PCs). Using the mean values of PC1, PC2, and PC3 with the stimulated hind paw indicated in blue (top); the mean values of PC2 and PC3, while varying PC1 either side of its mean by one standard deviation (middle-top); the mean values of PC1 and PC3, while varying PC2 (middle-bottom); and the mean values of PC1 and PC2, varying PC3 (bottom). (**D**) Behavioral trajectories of the 80 trials in PC space, showing 35–115 ms after stimulation. Only the first two PCs are shown for clarity. (**E**) PC vectors based on (**D**) show that trajectories are largely in the same direction. (**F**) The response magnitude (shown by colors that represent shift in PC2) varies as a function the initial pose, reduced to the first two PCs. (**G**) The initial PC values correlate with the shift in PC2 (left three plots). The initial PC3 value also correlates with the shift in PC3 (right). Least-squares linear fits are shown in blue and *r* values are Pearson's correlation coefficients. (**H**) Raster plots of the distances that each tracked body part moves relative to baseline in 80 trials from 10 mice. All raster plots are sorted by maximum distances achieved by the stimulated paw within 300 ms of the stimulation. (**I**) Six representative traces showing the Euclidean distance between the stimulated paw and nose. (**J**) This expansion and shortening of Euclidean distance between the stimulated paw and the nose are shown up to 300 ms post-stimulus for all 80 trials by plotting the maximum distances as a function of the minimum distance. Corresponding rug plots (orange ticks) and a kernel density estimate (gray lines) are shown. (**K**) Traces showing the angle of the nose normalized to mean baseline angle between the nose and tail base. The tail base reflects the origin in these calculations. 80 trials are shown, with stimulation on the left hind paw and right hind paw (top). Average traces are shown in *blue* and *red* for left and right hind paw stimulations, respectively. Polar histograms for mean nose yaw during 300 ms post-stimulus, corresponding to the traces directly above (below). The dataset is provided in *Figure 4—source data 2*.

The online version of this article includes the following video and figure supplement(s) for figure 4:

**Source data 1.** Whole-body motion energy recorded at 40 frames/s with different size stimuli.

**Source data 2.** Time courses for coordinates of six tracked body parts recorded at 400 frames/s.

**Figure supplement 1.** Motion energy analysis of behavior evoked by precisely controlled nociceptive input size.

**Figure supplement 2.** Motion energy analysis of high-speed recordings.

**Figure supplement 3.** Principal component analysis of shuffled behavioral data.

**Figure 4—video 1.** Markerless tracking of behavior in response to nociceptive stimulation.

https://elifesciences.org/articles/62026/figures#fig4video1

about the stimulus or its consequences, and potentially provides coping strategies. Protective behaviors can be statistically categorized (*Abdus-Saboor et al., 2019*) and computational discrimination of high-speed hind paw responses used as a score of pain (*Jones et al., 2020*). We have shown that the analysis can easily be customized to incorporate computational tools that facilitate quantification and reveal insights into complex behavioral responses.

## Behavioral responses to precise LTMR input

The vesicular glutamate transporter-1 (Vglut1) is a known marker of Aβ-LTMRs (*Alvarez et al., 2004*; *Brumovsky et al., 2007*). To demonstrate the utility of the system in the broader context of

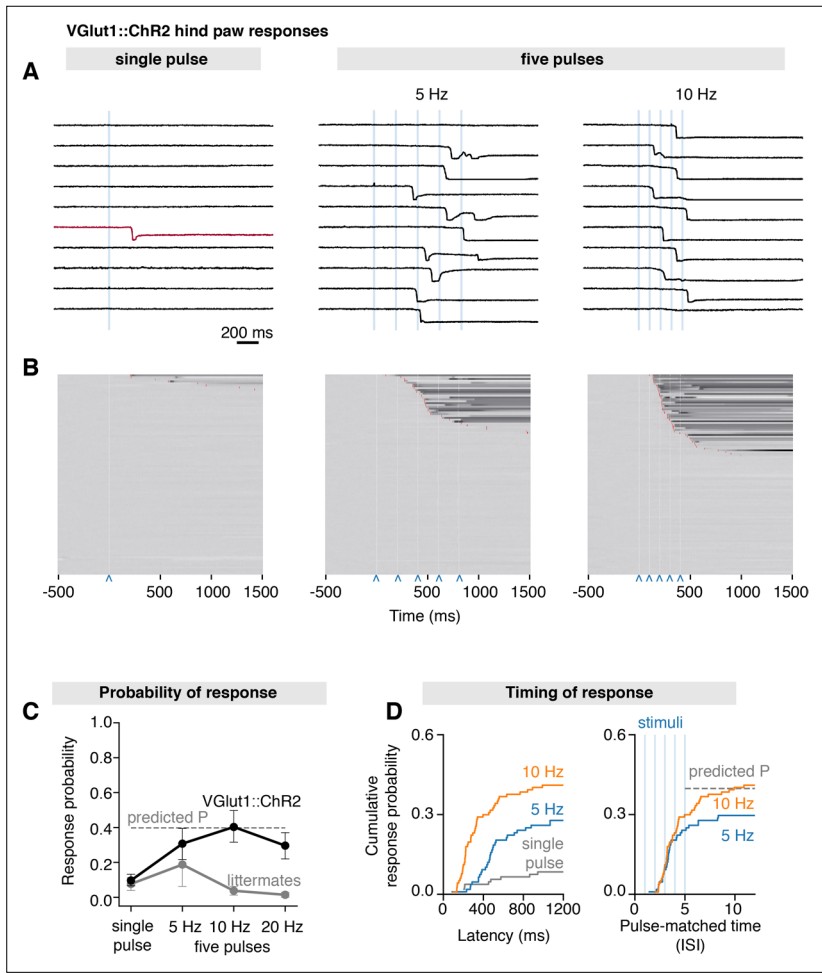

**Figure 5.** Scanned transdermal optogenetic activation of Aβ-LTMRs triggers slow-onset responses. (**A**) Example traces of the near-infrared frustrated total internal reflection signal time course for three different stimulation protocols in Vglut1^Cre::ChR2 mice: single pulse, five pulses at 5 Hz, and five pulses at 10 Hz (pulse duration 3 ms, spot size $S_7$ = 1.155 mm²). (**B**) Corresponding raster plots of hind paw responses sorted by latency. The paw response latency is indicated in red (99–103 trials/protocol from n = 11–12 mice) and the 3 ms laser stimuli shown with blue carets. (**C**) Paw response probability peaks at 10 Hz stimulation frequency in Vglut1^Cre::ChR2 mice (pulse duration 3 ms, spot size $S_7$ = 1.155 mm²; 99–103 trials/protocol from n = 11–12 mice, mean probability ± SEM). (**D**) Left panel: paw response latencies in trials with a single 3 ms stimulation or with trains of five 3 ms stimuli at 5 Hz or at 10 Hz. Right panel: paw response latencies normalized to the interstimulus interval. The estimated probability in (**C**) and (**D**) (dashed gray lines) was calculated using $P(X \geq 1) = 1-(1-p)^n$, where $p$ is the probability of a response on a single pulse (0.096) and $n$ is the number of pulses (5). The dataset is provided in *Figure 5—source data 1*.

The online version of this article includes the following figure supplement(s) for figure 5:

**Source data 1.** Time courses of paw movement recorded at 1000 frames/s with stimuli that vary in frequency.

**Source data 2.** Whole-body motion energy recorded at 400 frames/s with stimuli that vary in frequency.

**Figure supplement 1.** Motion energy analysis of full-body behavior-evoked Aβ-LTMRs.

somatosensation, we crossed *Slc17a7*-IRES2-Cre-D (Vglut1[Cre]) mice with R26-CAG-LSL-ChR2-tdTomato mice to generate a line (Vglut1[Cre]::ChR2) that express ChR2 in LTMRs (*Harris et al., 2014*). A recent detailed anatomical and physiological characterization of Vglut1[Cre]::ChR2 mice further confirmed that in DRG neurons, ChR2 is restricted to broad class of myelinated Aβ-LTMRs (*Chamessian et al., 2019*). Here, we find that a single 3 ms stimulus ($S_7$ = 1.155 mm$^2$) precisely delivered to the hind paw of these mice rarely elicited hind paw responses (mean paw withdrawal probability = 0.10 ± 0.03 SEM, 99 trials from n = 11 mice), with the earliest response occurring at 206 ms after stimulation (*Figure 5A and B*), which is an order of magnitude slower than we observed in TRPV1[Cre]::ChR2 mice (fastest response: 19 ms). Trains of five pulses, however, frequently elicited responses, showing mean paw withdrawal probabilities of 0.31 ± 0.09 (SEM, 108 trials from n = 12 mice) for 5 Hz and 0.40 ± 0.10 (SEM, 117 trials from n = 12 mice) for 10 Hz trains (*Figure 5C*). Increasing stimulation frequency to 20 Hz did not result in higher withdrawal probabilities, which may reflect ChR2 desensitization, rather than a physiological process (*Lin, 2011*). While the responses at first seem to be frequency-dependent (*Figure 5D*, left), inspection of recordings indicated that these occurred after the second or third pulse in most trials, regardless of stimulation frequency (*Figure 5A*). We find that the response distributions superimpose when withdrawal latencies are normalized to the interstimulus interval (pulse-matched latencies in *Figure 5D*, right). This observation suggests that response probability is likely driven by pulse summation, rather than by stimulation frequency. Indeed, we find that the probabilities and latencies can be explained by the probability sum rule, using the values for a single pulse to predict the values for five pulses (*Figure 5C and D*).

The magnitude of whole-body motion was not altered by increasing frequencies (*Figure 5—figure supplement 1*). In contrast to the TRPV1[Cre]::ChR2 line, whole-body behaviors in response to optogenetic stimulation of Vglut1[Cre]::ChR2 mice were subtle: visual inspection of high-speed whole-body behavior videos revealed that responses were mostly limited to small hind paw lifts or shifts towards the center of the body in cases where the stimulated paw was initially further away from the body. In most instances, these movements did not disturb balance or alter the animal's posture. Interestingly, we observed that whisking and, to a lesser extent, circular movements of the upheld forepaws would precede hind paw responses and initiate as early as the first pulse, even in trials that would not proceed to withdrawal. We speculate that mice may perceive the stimulation early on, but only act on this after a delay.

## Discussion

We describe a strategy for remote, precise, dynamic somatosensory input and behavioral mapping in awake unrestrained mice. The approach can remotely deliver spatiotemporally accurate optogenetic stimuli to the skin with predefined size, geometry, duration, timing, and location, while simultaneously monitoring behavior in the millisecond timescale. Microscale optogenetic stimulation can be used to simulate patterns, edges, and moving points on the skin. Responses to these precisely defined points and patterns can be mapped using machine vision approaches. The design is modular, for example, additional lasers for multicolor optogenetic control or naturalistic infrared stimuli can be added and complementary machine vision analysis approaches readily implemented. As an example, we combine this with DeepLabCut (*Mathis et al., 2018*), for markerless tracking of individual body parts to further dissect specific components of whole-body responses.

We validated the system in two transgenic mouse lines, providing optical control of broad-class Aδ and C-nociceptors, and Aβ-LTMRs. Advances in transcriptional profiling have identified a vast array of genetically defined primary afferent neuron populations involved in specific aspects of temperature, mechanical, and itch sensation (*Usoskin et al., 2015*). Selective activation of these populations is expected to recruit a specific combination of downstream cells and circuits depending on their function. For example, nociceptive input generates immediate sensorimotor responses and also pain that acts as a teaching signal. This strategy can be thus combined with techniques to modify genes, manipulate cells and neural circuits, and record neural activity in freely behaving mice to probe these mechanisms (*Boyden et al., 2005*; *Kim et al., 2017*). We provide approaches to map behavioral responses to defined afferent inputs across the spectrum of somatosensory modalities (*Browne et al., 2017*; *Huang et al., 2019*).

We find that the probabilistic recruitment of nociceptors determines the behavioral response probability, latency, and magnitude. We propose that the aggregate number of first action potentials

arriving from nociceptors to the spinal cord can be utilized to optimize the timing and extent of rapid protective responses. These first action potentials could be summated by spinal neurons so that appropriate behaviors are selected based on thresholds. Resultant fast behaviors are diverse but include coordinated head orientation and body repositioning that depends on the initial pose. In contrast, responses to optogenetic activation of Aβ-LTMRs occurred with slower onset, lower probability, and resulted in more subtle whole-body movements. Using a fixed number of pulses, we find that responses from multiple Aβ-LTMR inputs can be explained by the sum rule of probabilities rather than frequency-dependence (*Chamessian et al., 2019*). This does not, however, rule out the tuning of responses to more spatially or temporally complex stimuli. We used broad-class Cre driver lines to selectively stimulate either nociceptors or Aβ-LTMRs, and it is possible that their respective subpopulations exploit a diversity of coding strategies. This optical approach can reveal how such subpopulation and their specific downstream circuits guide behavior.

In summary, we have developed a strategy to precisely control afferents in the skin without touching or approaching them by projecting light to optogenetically generate somatosensory input in patterns, lines, or points. This is carried out non-invasively in awake freely behaving mice in a way that is remote yet precise. Remote control of temporally and spatially precise input addresses the many limitations of manually applied contact stimuli. The timing, extent, directionality, and coordination of resultant millisecond-timescale behavioral responses can be investigated computationally with specific sensory inputs. This provides a way to map behavioral responses, circuits, and cells recruited by defined afferent inputs and dissect the neural basis of processes associated with pain and touch. This strategy thus enables the investigation of sensorimotor, perceptual, cognitive, and motivational processes that guide and shape behavior in health and disease.

# Materials and methods

**Key resources table**

| Reagent type (species) or resource | Designation | Source or reference | Identifiers | Additional information |
|---|---|---|---|---|
| Genetic reagent (*Mus musculus*) | R26-CAG-LSL-hChR2(H134R)-tdTomato (Ai27D) | Jackson Laboratory | Stock #: 012567 RRID: IMSR_JAX:012567 | PMID:22446880 |
| Genetic reagent (*M. musculus*) | *Trpv1*-IRES-Cre (TRPV1^Cre) | Jackson Laboratory | Stock #: 017769 RRID: IMSR_JAX:017769 | PMID:21752988 |
| Genetic reagent (*M. musculus*) | *Slc17a7*-IRES2-Cre-D (Vglut1^Cre) | Jackson Laboratory | Stock #: 023527 RRID: IMSR_JAX:023527 | PMID: 21752988 |
| Software, algorithm | RStudio | RStudio http://www.rstudio.com/ | RRID:SCR_000432 | Version 1.2.5019 |
| Software, algorithm | Python | Python http://www.python.org/ | RRID:SCR_008394 | Version 3.6.8 |
| Software, algorithm | Fiji | Fiji http://fiji.sc | RRID:SCR_002285 | Version 2.0.0 |
| Software, algorithm | Prism 7 | GraphPad Prism http://www.graphpad.com/ | RRID:SCR_002798 | Version 7 |
| Software, algorithm | Seaborn | Seaborn http://www.seaborn.pydata.org | RRID:SCR_018132 | |
| Software, algorithm | Adobe Illustrator | Adobe http://www.adobe.com | RRID:SCR_010279 | Version 24.0 |

## Optical system design, components, and assembly

Optical elements, optomechanical components, mirror galvanometers, the diode laser, LEDs, controllers, machine vision cameras, and structural parts for the optical platform are listed in the table in *Supplementary file 1*. These components were assembled on an aluminum breadboard as shown in the Solidworks rendering in *Figure 1C*. The laser was aligned to the center of all lenses and exiting the midpoint of the mirror galvanometer housing aperture when the mirrors were set to the center of their working range. A series of lenses (L1–L3) expanded the beam before focusing it on to the glass stimulation plane, on which mice are placed during experiments. The glass stimulation platform was constructed of 5- mm-thick borosilicate glass framed by aluminum extrusions. NIR-FTIR

was achieved by embedding an infrared LED ribbon inside the aluminum frame adjacent to the glass edges (Roberson, D. P. et al., manuscript submitted). The non-rotating L1 lens housing was calibrated to obtain eight defined laser spot sizes, ranging from 0.0185 mm$^2$ to 2.307 mm$^2$, by translating this lens along the beam path at set points to defocus the laser spot at the 200 mm × 200 mm stimulation plane. The beam size can be altered manually using this rotating lens tube per design, but this is modular and could be altered by the user. To ensure a relatively flat field in the stimulation plane, the galvanometer housing aperture was placed at a distance of 400 mm from its center. In this configuration, the corners of the stimulation plane were at a distance of 424 mm from the galvanometer housing aperture and variability of the focal length was below 1.5 %.

Optical power density was kept constant by altering the laser power according to the laser spot area. Neutral density (ND) filters were used so that the power at the laser aperture was above a minimum working value (≥8 mW) and to minimize potential changes in the beam profile at the stimulation plane. The laser and mirror galvanometers were controlled through a multifunction DAQ (National Instruments, USB-6211) using custom software written in LabVIEW. The software displays the NIR-FTIR camera feed, whose path through the mirror galvanometers is shared with the laser beam, so that they are always in alignment with one another. Computationally adjusting mirror galvanometer angles causes identical shifts in both the descanned NIR-FTIR image field of view and intended laser stimulation site, so that the laser can be targeted to user-identified locations. Shaped stimulation patterns were achieved by programmatically scaling the mirror galvanometer angles to the glass stimulation plane using a calibration grid array (Thorlabs, R1L3S3P). The timings of laser pulse trains were synchronized with the mirror galvanometers to computationally implement predefined shapes and lines using small angle steps that could be as short as 300 μs. The custom software also synchronized image acquisition from the two cameras, so that time-locked high-speed local paw responses were recorded (camera 1: 160 pixels × 160 pixels, 250–1000 frames/s depending on the experiment). Time-locked global whole-body responses were recorded above video-frame rate (camera 2: 664 pixels × 660 pixels, 40 frames/s) or at high speed (camera 2: 560 pixels × 540 pixels, 400 frames/s) across the entire stimulation platform.

## Technical calibration and characterization of the optical system

To calibrate the L1 lens housing and ensure consistency of laser spot sizes across the glass stimulation platform, we designed a 13.90 ± 0.05 mm thick aluminum alignment mask. This flat aluminum mask was used to replace the glass stimulation platform and was combined with custom acrylic plates that align the aperture of a rotating scanning-slit optical beam profiler (Thorlabs, BP209-VIS/M) to nine defined coordinates at different locations covering the stimulation plane. The laser power was set to a value that approximates powers used in behavioral experiments (40 mW). The laser power was then attenuated with an ND filter to match the operating range of the beam profiler. Using Thorlabs Beam Software, Gaussian fits were used to determine x-axis and y-axis 1/e$^2$ diameters and ellipticities for each laser spot size over three replicates at all nine coordinates. The averages of replicates were used to calculate the area of the eight different laser spot sizes that were measured in each of the nine coordinates (*Figure 1—figure supplement 1A*) and then fitted with a two-dimensional polynomial equation in MATLAB to create heatmaps (*Figure 1—figure supplement 1* B).

The average values over the nine coordinates were defined for each laser spot size: $S_1$ = 0.0185 mm$^2$, $S_2$ = 0.0416 mm$^2$, $S_3$ = 0.0898 mm$^2$, $S_4$ = 0.176 mm$^2$, $S_5$ = 0.308 mm$^2$, $S_6$ = 0.577 mm$^2$, $S_7$ = 1.155 mm$^2$, $S_8$ = 2.307 mm$^2$. These measurements were repeated 6 months after extensive use of the optical system to ensure stability over time (*Figure 1—figure supplement 1A*). In addition, the uniformity of laser power was assessed by measuring optical power at five positions of the experimental platform with a power meter (Thorlabs, PM100D) (*Figure 1—figure supplement 1C*).

## Experimental animals

Experiments were performed using mice on a C57BL/6 j background. Targeted expression of ChR2-tdTomato in broad-class cutaneous nociceptors was achieved by breeding mice homozygous for Cre-dependent ChR2(H134R)-tdTomato at the Rosa26 locus (RRID:IMSR_JAX:012567, R26-CAG-LSL-hChR2(H134R)-tdTomato, Ai27D; *Madisen et al., 2012*) with mice that have Cre recombinase inserted downstream of the *Trpv1* gene in one allele (RRID:IMSR_JAX:017769, *Trpv1*-IRES-Cre, TRPV1[Cre]; *Cavanaugh et al., 2011*). Aβ-LTMRs were selectively stimulated by breeding homozygous Ai27D mice with

mice in which Cre recombinase is targeted to cells expressing the vesicular glutamate transporter 1 (RRID:IMSR_JAX: 023527, Slc17a7-IRES2-Cre-D, Vglut1$^{Cre}$; *Harris et al., 2014*). Resultant mice were heterozygous for both transgenes and were housed with control littermates that do not encode Cre recombinase but do encode Cre-dependent ChR2-tdTomato. Adult (2–4 months old) male and female mice were used in experiments. Mice were given ad libitum access to food and water and were housed in 21°C ± 2°C, 55 % relative humidity and a 12 hr light:12 hr dark cycle. Experiments were carried out on at least two separate cohorts of mice, each cohort contained 4–6 mice. Experiments were spaced by at least one day in the case where the same cohort of mice was used in different experiments. All animal procedures were approved by University College London ethical review committees and conformed to UK Home Office regulations.

## Optogenetic stimulation and resultant behaviors

Prior to the first experimental day, mice underwent two habituation sessions during which each mouse was individually placed in a plexiglass chamber (100 mm × 100 mm, 130 mm tall) on a mesh wire floor for 1 hr, then on a glass platform for another hour. On the experimental day, mice were again placed on the mesh floor for 1 hr, then up to six mice were transferred to six enclosures (95 mm × 60 mm, 75 mm tall) positioned on the 200 mm × 200 mm glass stimulation platform. Mice were allowed to settle down and care was taken to stimulate mice that were calm, still, and awake in an 'idle' state. The laser was remotely targeted to the hind paw glabrous skin using the descanned NIR-FTIR image feed. The laser spot size was manually set using the calibrated L1 housing, while laser power and neutral density filters were used to achieve a power density of 40 mW/mm$^2$ regardless of spot size. The software was then employed to trigger a laser pulse of defined duration (between 100 µs and 30 ms) and simultaneously acquire high-speed (1000, 500, or 250 frames/s depending on experiment) NIR-FTIR recordings of the stimulated paw, as well as a global view of the mice with a second camera (400 frames/s or 40 frames/s) (*Figure 1C*). Recordings of stimulations of TRPV1$^{Cre}$::ChR2 mice were 1500 ms in duration, with the laser pulse initiated at 500 ms. For each stimulation protocol, six pulses, three on each hind paw, spaced by at least 1 min were delivered to eight mice, split into two cohorts. For experiments involving Vglut1$^{Cre}$::ChR2 mice, we used a single stimulation spot size ($S_7$ = 1.155 mm$^2$) and duration (3 ms). In addition to the single-pulse stimulation, these mice received a train of five pulses applied at 5, 10, or 20 Hz. The recording time for each trial was extended to 2000 ms to accommodate for the longer stimulation period. For each protocol, Vglut1$^{Cre}$::ChR2 mice were stimulated in 10 trials, split equally between the two hind paws. Data was collected from 12 Vglut1$^{Cre}$::ChR2 mice and 8 littermate controls lacking Cre recombinase split into five cohorts. In all experiments, the behavioral withdrawal of the stimulated hind paw was also manually recorded by the experimenter.

## Patterned stimulation protocols

TRPV1$^{Cre}$::ChR2 mice were stimulated on the heel of the hind paw with each of the following protocols: (1) a single 1 ms pulse with spot size $S_7$ (1.155 mm$^2$); (2) a single 1 ms pulse with spot size $S_4$ (0.176 mm$^2$); (3) seven 1 ms pulses with spot size $S_4$, superimposed on the same stimulation site and spaced by 500 µs intervals; (4) seven 1 ms pulses with spot size $S_4$, spaced by 500 µs intervals and spatially displacing stimuli with 0.3791 mm jumps such as to draw a small hexagon; (5) seven 1 ms pulses with spot size $S_4$, spaced by 500 µs intervals and spatially displacing stimuli with 0.5687 mm jumps such as to draw a hexagon expanded by 50 % compared to the previous shape; and (6) seven 1 ms pulses with spot size $S_4$, spaced by 500 µs intervals and spatially displacing stimuli with 0.3791 mm jumps such as to draw a straight line. The power density of the stimulations was kept constant at 40 mW/mm$^2$ as before. Seven mice, split into two cohorts, received 10 stimulations per protocol (five on each hind paw) after a baseline epoch of 500 ms. An additional cohort of four littermates lacking Cre recombinase were stimulated in the same way and served as negative controls. Finally, three TRPV1$^{Cre}$::ChR2 mice were stimulated (spot size $S_8$, 10 ms pulse duration) with a single pulse adjacent to the hind paw, five times on each side, in order to control for potential off-target effects. The NIR-FTIR signal was recorded at 500 frames/s.

## Whole-body behaviors during optogenetic stimulation

To obtain recordings optimized for markerless tracking with DeepLabCut, a single acrylic chamber (100 mm × 100 mm, 150 mm tall) was centered on the glass stimulation platform of the system. Rapid

movements were recorded at 400 frames/s using a below-view camera (FLIR, BFS-U3-04S2M-CS). Two white and two infrared LED panels illuminated the sides of the behavioral chamber in order to optimize lighting for these short exposure times and achieve high contrast images. NIR-FTIR was not used in this configuration. TRPV1$^{Cre}$::ChR2 mice received between 10 and 20 single-shot laser pulse stimulations of 10 ms each, at least 1 min apart and equally split between right and left hind paw and using spot size $S_8$ (2.31 mm$^2$). The first 10 trials that exceeded quality control were used (see *Markerless tracking of millisecond-timescale global behaviors, Data processing*). Each trial consisted of a 500 ms baseline and 4000 ms after-stimulus recording epoch.

## Automated analysis of optogenetically evoked local withdrawal events

High-speed NIR-FTIR recordings were saved as uncompressed AVI files. A Python script was implemented in Fiji to verify the integrity of the high-speed NIR-FTIR recordings and extract average 8-bit intensity values from all frames within a circular region of interest on the stimulation site (60 pixels diameter). This output was then fed into RStudio to calculate the average intensity and associated standard deviation of the baseline recording (first 500 ms). A hind paw response was defined as a drop of intensity equal to or below the mean of the baseline minus five times its standard deviation. Paw response latency was defined as time between the start of the pulse and the time at which a hind paw response was first detected. For purposes of quality control, only recordings with a baseline NIR-FTIR intensity mean ≥ 3 and a standard deviation/mean of the baseline ratio ≥23 were retained for analysis. Another criterion was that response latencies are not 10 ms or shorter since this would be too short to be generated by the stimulus itself. Only one trial out of 2369 trials did not meet this criterion (spot size $S_6$, 1 ms pulse, 8 ms response latency). In addition to this two-step workflow using Fiji/Python to process AVI files and then RStudio to analyze the resulting output, alternative code was written in Python 3, which combines both steps and also computes individual pixel latencies and motion energy using NumPy and Pandas packages. A median filter (radius = 2 pixels) was applied to the NIR-FTIR recordings used to create the representative time series in *Figure 2A* and *Figure 2— video 1*. For raster plots of hind paw response dynamics in *Figure 4A*, NIR-FTIR intensity values were normalized to the average baseline value. For the patterned stimulation experiments in *Figure 2F* and Vglut1$^{Cre}$::ChR2 experiments in *Figure 5A–D*, trials were analyzed as stated to compute local response probabilities, but an additional rule was introduced to further minimize the risk of false positives. A response required the signal to fall by 20 % and exceed a threshold of four times the standard deviation of baseline. Compared to the performance of an experimenter manually processing the videos with Fiji, the automated analysis pipeline was substantially faster for similar accuracy. For example, it took an experimenter two working days to analyze 127 videos, whereas the Fiji/Python pipeline generated the identical output within 90 s.

## Automated analysis of whole-body protective behavior

Videos of the entire stimulation platform were cropped into individual mouse chambers (200 × 315 pixels) and then analyzed using RStudio to quantify the amount of whole-body movements, including those stemming from the response of the stimulated limb, herein referred to as global behavior (GB). GB was approximated as the binarized motion energy: the summed number of pixels changing by more than five 8-bit values between two subsequent frames (Pixel Change). Briefly, for each pixel$_n$ (n = 63,000 pixels/frame), the 8-bit value at a given frame ($F_n$) was subtracted from the corresponding pixel$_n$ at the previous frame ($F_{n-1}$). If the resulting absolute value was ≤5, 0 would be assigned to the pixel. If the absolute resulting value was >5, 1 would be assigned to the pixel. The threshold was chosen to discard background noise from the recording. The pixel binary values were then summed for each frame pair to obtain binarized motion energy. Normalized binarized motion energy was calculated by subtracting each post-stimulus frame binarized motion energy from the average baseline binarized motion energy. As an alternative to this analysis strategy, we have developed code in Python that processes the video files and calculates motion energy. The peak normalized binarized motion energy was determined and only trials displaying a peak response ≥5 standard deviations of the baseline mean were retained for further analysis and plotting. For TRPV1$^{Cre}$::ChR2 mice, the analysis was restricted to a time window of 100 ms after stimulus onset (first three frame pairs proceeding the stimulus frame) to enable time-locking to the stimulus. Between 41 and 47 videos from eight mice were analyzed per spot size. For experiments with Vglut1$^{Cre}$::ChR2 mice, the peak normalized binary

motion energy exceeding five standard deviations of the baseline mean was determined for the entire 1.5 s recording epoch proceeding stimulus onset. Between 51 and 80 trials from 11 to 12 mice were analyzed per stimulation frequency.

## Markerless tracking of millisecond-timescale global behaviors

### DeepLabCut installation

DeepLabCut (version 2.0.1) was installed on a computer (Intel-Core-i7-7800 × 3.5 GHz CPU, NVIDIA GTX GeForce 1080 Ti GPU, quad-core 64 GB RAM, Windows 10, manufactured by PC Specialist Ltd.) with an Anaconda virtual environment and was coupled to Tensorflow-GPU (v.1.8.0, with CUDA v.9.01 and cUdNN v. 5.4).

### Data compression

All recordings were automatically cropped with Python MoviePy package and compressed with standard compression using the H.264 format, then saved in mp4 format. This compression method was previously shown to result in robust improvement of processing rate with minimal compromise on detection error.

### Training the network

DeepLabCut was used with default network and training settings. Pilot stimulation trials were collected for initial training with 1,030,000 iterations from 253 labeled images from 50 videos. The videos were selected to represent the whole range of behavioral responses and conditions (25 videos of males and 25 videos of females from six different recording sessions). Out of the 25 videos, 15 were selected from the most vigorous responses, 5 were selected from less vigorous responses, and 5 from control mice. Ground truth images were selected manually, aiming to include the most variable images from each video (up to 14 frames per video). 18 body parts were labeled, namely the nose, approximate center of the mouse, two points on each sides of the torso and one point at each side of the neck, the fore paws, distal and proximal points on the hind paw, between the hind limbs, and three points on the tail. While most of these labels were not used in subsequent analysis, labeling more body parts on the image enhanced performance. The resulting network output was visually assessed. Erroneously labeled frames were manually corrected and used to retrain the network while also adding new recordings. Four sequential retraining sessions with 1,030,000 iterations each were conducted adding a total of 109 frames from 38 videos. This resulted in a reduction in the pixel root mean square error (RMSE) from 4.97 down to 2.66 on the test set, which is comparable to human ground truth variability quantified elsewhere.

### Data processing

Only labels of interest were used for analysis. These were ipsilateral and contralateral hind paws (distal), the tail base, and the nose labels. To minimize error, points were removed if they (1) were labeled with less than 0.95 p-cutoff confidence by DeepLabCut, (2) jumped at least 10 pixels in one single frame compared to the previous frame, (3) had not returned on the subsequent frame, and (4) were from the five stimulation frames. Code for data processing was written in Python using the NumPy and Pandas packages. Additional post-hoc quality control was performed on the network output to identify and remove poorly labeled trials. To this end, heatmaps of distances between labels were created and inspected for dropped labels and sudden changes in distance. Trials identified in this manner were then manually inspected and removed if more than 10 % of labels were missing or more than 10 frames were mislabeled. In total, 4.7 % of trials were discarded. Only the first eight trials for each of the 10 mice that met this video quality control were used in analysis.

### Automated detection of the stimulated limb

Disabling NIR-FTIR illumination reduces the baseline saturation and thus allowed us to automate stimulated paw detection using pixel saturation from the stimulation laser. To determine which of the left or right paw had been stimulated in a given trial, the number of saturated pixels within a 60 × 60 pixels window close to the hind paw label was compared 7.5 ms prior and 5 ms after stimulus onset.

### Detection of movement latency of discrete body parts

Movement latencies of hind paws and head (nose) were computed based on significant changes from the baseline position. Baseline positions were calculated as the average x and y values from 10 consecutive frames prior to stimulus onset. A post-stimulus response was considered to be meaningful if the position of the label changed by at least 0.5 pixels (~0.16 mm) compared to baseline and continued moving at a rate of at least 0.5 pixel/frame for the subsequent 10 frames.

### Dimensionality reduction

We carried out dimensionality reduction on x and y values for six body parts (nose, left hind paw digits, left hind paw heel, right hind paw digits, right hind paw heel, and tail base) determined at a single time point. These were egocentrically aligned using the tail base as the origin, and the stimulated paw always on the right. PCA was carried out by extracting the first three PCs using these 12 features at 115 ms after stimulus onset. The PCA was cross-validated by pseudo-randomly splitting the 80 trials into training and test datasets (80:20). The training dataset showed 49.5, 27.4, and 12.3% variance was explained by PC1, PC2, and PC3, respectively. The same PCs explained 53.5, 23.2, and 10.1% variance in the test dataset. PCA of these 80 trials together (at 115 ms) gave explained variance values 50.4% (PC1), 26.5% (PC2), and 11.9% (PC3). Projecting the time courses onto these same PCs resulted in explained variance values 37.1% (PC1), 24.3% (PC2), and 16.7% (PC3). In all cases, the shifts seen in PC1–3 were similar to that shown in *Figure 4C*.

## Motion energy calculations in millisecond-timescale global behaviors

GB was analyzed within a 1 ms time frame following stimulation by computing the binarized motion energy relative to a baseline reference frame 5 ms prior to stimulation as described above. Here, the threshold for pixel change was set to seven 8-bit values. The binarized motion energy (sum of pixel binaries) of a given frame was normalized to the total number of pixels within that frame after removing those frames that had been affected by the stimulation laser pulse. The global response latency of movement initiation was determined as the time when binarized motion energy was greater than 10 times the standard deviation at baseline. Termination of movement was determined as the time point when binarized motion energy returned below 10 times standard deviation from baseline following the first movement bout.

## Statistical analysis

Data was analyzed in RStudio 1.2.5019, Python 3.6.8, ImageJ/Fiji 2.0.0 and Prism 7 and visualized using Seaborn, Prism 7, and Adobe Illustrator 24.0. In all experiments, repeated measurements were taken from multiple mice. Paw responses to patterned stimulation were reported as mean probabilities ± standard error of the mean (SEM) and analyzed using Friedman's non-parametric test for within-subject repeated measures followed by Dunn's signed-rank test for multiple comparisons (*Figure 2F*). In this experiment, one of the seven TRPV1$^{Cre}$::ChR2 mice was removed from the dataset because it displayed saturating responses to Protocol 3 preventing comparison of values across a dynamic range. Response latencies, response rise times, and response durations were computed using a hierarchical bootstrap procedure (*Saravanan et al., 2020*) modified to acquire bootstrap estimates of the median with balanced resampling. Briefly, mice are sampled with replacement for the number of times that there are mice. For each mouse within this sample, its trials were sampled with replacement, but the number of selected trials was balanced, ensuring each mouse contributes equally to the number of trials in the sample. The median was taken for this resampled population and this entire process was repeated 10,000 times. Bootstrap estimates from 1000 simulated experiments show that an additional 1.6–3.1% of values fall within 1 % of the population median for seven mice with between 2 and 6 responses. Values provided are the mean bootstrap estimate of the median ± the standard error of this estimate. The median bias was small due to the resampled population size from hierarchically nested data and only moderate distribution skew. Global peak motion energy (*Figure 4B*) was examined in a similar way, except the mean of resampled populations was used as it represents a better estimator of the population mean. In this case, we report the mean bootstrap estimate of the mean ± the standard error of this estimate. Pearson's correlation coefficients were determined to compare maximum distances moved from baseline for each body part (*Figure 4F*). Experimental units and n values are indicated in the figure legends.

## Acknowledgements

We are grateful to Dr Mehmet Fisek and Dr Adam M Packer for initial advice on the optical system, and thank Dr David P Roberson and the Woolf lab for sharing the NIR-FTIR technology. We gratefully acknowledge feedback on the manuscript from Dr Adam M Packer and Professor John N Wood. This work was supported by a Sir Henry Dale Fellowship jointly funded by the Wellcome Trust and the Royal Society (109372/Z/15/Z).

## Additional information

### Funding

| Funder | Grant reference number | Author |
| --- | --- | --- |
| Wellcome Trust | Sir Henry Dale Fellowship 109372/Z/15/Z | Liam E Browne |
| Royal Society | Sir Henry Dale Fellowship 109372/Z/15/Z | Liam E Browne |

The funders had no role in study design, data collection and interpretation, or the decision to submit the work for publication.

### Author contributions

Ara Schorscher-Petcu, Data curation, Formal analysis, Investigation, Methodology, Software, Validation, Visualization, Writing – original draft, Writing – review and editing; Flóra Takács, Data curation, Formal analysis, Investigation, Methodology, Software, Visualization, Writing – review and editing; Liam E Browne, Conceptualization, Data curation, Formal analysis, Funding acquisition, Investigation, Methodology, Project administration, Resources, Software, Supervision, Validation, Visualization, Writing – original draft, Writing – review and editing

### Author ORCIDs

Ara Schorscher-Petcu http://orcid.org/0000-0001-5808-5172
Liam E Browne http://orcid.org/0000-0002-5693-7703

### Ethics

All animal procedures were approved by University College London ethical review committees and conformed to UK Home Office regulations.

### Decision letter and Author response

Decision letter https://doi.org/10.7554/eLife.62026.sa1
Author response https://doi.org/10.7554/eLife.62026.sa2

## Additional files

### Supplementary files

• Supplementary file 1. List of components for the assembly of the optical system. List of parts used in system. A Solidworks assembly, the optical system control and acquisition software, and behavioral analysis toolkit are available at https://github.com/browne-lab/throwinglight (*Schorscher-Petcu and Browne, 2020*).

• Transparent reporting form

### Data availability

All components necessary to assemble the optical system are listed in Supplementary file 1. A Solidworks assembly, the optical system control and acquisition software and behavioral analysis toolkit are available at https://github.com/browne-lab/throwinglight. The data that support the findings of this study are provided as source data files.

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
