## [Decision Letter]

**Acceptance summary:**

We acknowledge the effort and time the authors took to address all of the comments made by reviewers which have improved this paper. We strongly believe that this approach will be widely adopted in the somatosensory research community to deliver "remote optogenetic touch" to freely behaving animals in a highly spatial and temporal fashion. The addition of a non-nociceptive line to this paper, with distinct behavioral outputs from the nociceptive line, is an added major strength of this work.

**Decision letter after peer review:**

Thank you for submitting your article "Scanned optogenetic control of mammalian somatosensory input to map input-specific behavioral outputs" for consideration by *eLife*. Your article has been reviewed by 3 peer reviewers, one of whom is a member of our board of Reviewing Editors and the evaluation has been overseen by Andrew King as the Senior Editor. The reviewers have opted to remain anonymous.

The reviewers have discussed the reviews with one another and the Reviewing Editor has drafted this decision to help you prepare a revised submission.

The manuscript by Schorscher-Petcu is a very innovative study approaching an important problem in pain and somatosensory neuroscience – precise and remote delivery of sensory stimuli. This work is timely, and there are many clear applications to understanding peripheral somatosensory encoding using this strategy. The rationale for such a tool developed here is widely agreed upon in the field, and if others can easily adopt this strategy, this could become the standard for peripheral optogenetic stimulation of the hind paw. Nonetheless, the reviewers would like the authors to address the points listed here before consideration of publication in *eLife*.

1. The strength of this paper was in the technique and not the new biology uncovered. We would like the authors to remove any language of a "sparse code" proposition because the data presented here do not support such a claim.

2. More detail on how to use this system so that new users can use this off-the-shelf. In particular: we had a hard time evaluating the hierarchical bootstrap procedure, which references a pre-print. Is this method really ensuring that the results are more rigorous? How do the authors define a withdraw? More detail and commentary on how this approach interfaces with Deep Lab Cut. In general, focus more on the technique and less on biology.

3. For widespread applicability and to determine the range, strengths, and weaknesses of this new tool to the pain field, the authors should extend their behavioral analyses. The reviewers preferred the authors to do as they mention in the discussion, which is to add an additional somatosensory line (perhaps a non-pain line) and see how their platform performs in comparison to Trpv1-ChR2. The less preferred option if the authors are not able to breed new somatosensory lines in reasonable time, is to try the Trpv1-ChR2 line in different contexts (inflammatory and/or neuropathic pain). In either case, at baseline or during chronic pain states in the Trpv1-ChR2 line, the authors should use an analgesic and show that their tool is modular and can detect decreases in pain-related signatures. The authors should take care to have N numbers closer to 10 animals per group, as the N of 4 in their studies is on the lower side.

---

## [Author Response]

1. The strength of this paper was in the technique and not the new biology uncovered. We would like the authors to remove any language of a "sparse code" proposition because the data presented here do not support such a claim.

We thank the reviewers and agree that as a technical report we should focus on the technical aspects. In demonstrating the utility of the system, we reveal biological insights that should be described clearly and fully. The language used is critical, so based on the reviewers comments we have removed the term "sparse code". We agree that "code" unintentionally implied that we show noxious stimuli are encoded naturally in this manner in this combination of broad-class nociceptors. More precisely, we are reporting that the relative size of a single action potential nociceptor volley can determine the nature, timing and extent of resultant protective behaviors. This relationship between nociceptive input and behavior has not been demonstrated before and so it is important to describe this. We do however retain the use of the term "sparse" as this summarizes the optogenetic stimulus for the reasons below.

First, the input is spatially sparse – stimuli activate a small percentage of nociceptors in the total population. In Figure 2D, we show even an area as small as 0.6 mm² can tune the behavioral response. We now emphasize this point in the manuscript “The stimulus used for this protocol was S6, 0.577 mm² in area, which corresponds to less than 1% of the glabrous paw area and highlights the sensitivity of the nociceptive system”. Second, the input is temporally sparse – the stimuli only generate a single action potential in each nociceptor and this is enough to generate coordinated behaviors. That this is temporally sparse is in line with common usage of the term (e.g. Hahnloser et al., Nature 2002). This builds on our previous work that showed that a single pulse only generates a single action potential in each nociceptor.

We confirm that the subtle difference in laser rise times have no implications on generalizing the previous findings. Indeed, the TRPV1-lineage used here have long refractory periods (minimum interspike intervals, ~20 ms), as measured by patch clamp recordings (Browne et al., Cell Reports 2017). We had previously reported that even a 10 ms optogenetic stimulation of this mouse line does not generate more than a single action potential (see Browne et al., Cell Reports 2017 Supplementary Information, in vitro electrophysiology), but this is perhaps not prominent enough in the previous paper. In the system we describe here, the diode laser delivers the same wavelength and intensity of light as before. The light reaches a stable intensity in less than 1 microsecond, which allowed us to precisely compare the effects of short pulses as the ChR2 kinetics (tau=1.9 ms, Lin, 2011) and are what limits the rate of depolarization, not these lasers. Figure 2E uses 1 ms and 3 ms pulses only, and Figure 2F uses 1 ms pulses. We show that paw response probability increases from 0.3 ms up to 10 ms (Figure 2D), which is not long enough to generate more than one action potential. We now point out alongside Figure 2D in the legend that the 30 ms pulse duration may result in more than one action potential but it is important to note this does not alter the interpretation as the probability already plateaued at 10 ms.

Sparse recruitment of broad-class nociceptors was achieved by gaining precise control over the pulse width and spatial properties (size and shape) of the stimuli. It is not known whether certain pulse durations are more selective for the fibers they activate, so we have used a range of approaches to control stimulation size. We find that the larger the stimulus, irrespective of increasing the spot size or increasing the duration of the pulse, the shorter the response latencies. As illumination area increases so does the probability that Aδ and C fibres are activated. Our analysis of hind paw latencies was focused on Aδ responses. This is possible due to the stimuli being time-locked and the differential conduction velocities of these fibres – the responses typically initiate before 35 ms when the C-fiber action potentials (1.5 m/s) will only just be arriving at the spinal cord. We have previously shown using electrophysiology that TRPV1-Cre::ChR2 drives expression of ChR2 in broad-class nociceptors including Aδ and C fibers (Browne et al., Cell Reports, 2017). This is also in line with evidence from Usoskin et al., (Nature Neuroscience, 2015) showing the PEP2 population expresses TRPV1, and discussed by Caterina and Julius (Annual Review of Neuroscience, 2001) and Mitchell et al., (Mol Pain 2010).

2. More detail on how to use this system so that new users can use this off-the-shelf. In particular: we had a hard time evaluating the hierarchical bootstrap procedure, which references a pre-print. Is this method really ensuring that the results are more rigorous? How do the authors define a withdraw? More detail and commentary on how this approach interfaces with Deep Lab Cut. In general, focus more on the technique and less on biology.

We thank the reviewers and have made changes throughout. Specific responses and actions are given below.

Bootstrap. The bootstrap procedure allows statistical error to be calculated and provides better estimates of central tendency in skewed distributions. Here, we use hierarchical bootstrap, which as the now peer-reviewed article from Sober and colleagues points out (Saravanan et al., Neuron Behav Data Anal Theory, 2020), is an extensively used statistical method for nested data. We modified this to ensure that each mouse contributes equally to the number of trials in the sample, to avoid bias. By running our own analysis of this procedure, we confirm that the results are more rigorous; for 7 mice with between 2 and 6 responses, an additional 1.6-3.1% of bootstrap estimates from 1000 simulated experiments (with skewed distributions) fall within 1% of the population median. We have added this to the manuscript.

Targeting stimuli. The laser is fixed to the center of view in the targeting camera, which is displayed in real time in the GUI. This is due to the "descanned" configuration of the laser and targeting camera both using the same path through the galvanometers. Moving the target location of the laser moves the field of view that is displayed. The experimenter chooses what exact location of the mouse to stimulate by aiming the laser with this camera feed. We used a computer mouse cursor to rapidly target stimuli. On triggering a trial, the same camera records 0.5 seconds before the stimulus and 1.5 second after stimulus onset. These high-speed (up to 1000 fps) videos are used to verify if any movement occurred prior to the stimulation. There are occasions when the mouse moves before the stimulation is delivered, which is unavoidable in freely moving mice but the pre-stimulus baseline from all recordings are examined for movement using our code. In practical terms, we assess whether there has been movement occurring during the 0.5 seconds preceding the stimulus by calculating the mean signal intensity for this epoch and dividing it by its standard deviation. If the animal is still, the standard deviation will be small and the incurring ratio large. If the mouse moved, the standard deviation will be high and the ratio small. We empirically determined a threshold value, which we found to reliably indicate whether the animal is not moving up until the delivery of the stimulation. If there is any movement, the trial will not be used thus removing the possibility of recordings with unintended stimulus delivery.

To provide clarification on the targeting of stimuli, we have added further description to the Results section Design and assembly of the optical stimulation approach, which now states:

“The camera-feed is displayed in the user interface and enables the operator to use this image to target the laser to the desired location”.

This makes it clearer how the laser is remotely targeted. The reader may also refer to the Materials and methods section for a more detailed description:

“The software displays the NIR-FTIR camera feed, whose path through the mirror galvanometers is shared with the laser beam, so that they are always in alignment with one another. Computationally adjusting mirror galvanometer angles causes identical shifts in both the descanned NIR-FTIR image field of view and intended laser stimulation site, so that the laser can be targeted to user-identified locations.”

Paw responses. We define a hind paw response as a decrease in the intensity equal to or below the mean of the baseline minus five times its standard deviation. This is stated in Materials and methods section *Automated analysis of optogenetically evoked local withdrawal events*. By this definition, a paw response does not require the paw to lift / withdraw. We examine the extent of the response and can categorize partial and full responses. For example, if only part of the paw moves without lifting from the floor this is not a full paw withdrawal but still considered a response and it contains information that can be included in the analysis. The responses are detected using frustrated total internal reflection of infrared light at 1,000 frames per second. These resultant videos reveal nuance that would not be detected with the human eye. The videos are analyzed with code written in R and Python. We extract each of the 25,600 pixels for each frame and statistically analyze changes in each pixel or the intensity across a region of interest. The code is freely available. The videos are inspected by eye to validate findings. We have made changes in the manuscript that make this clearer.

Open science. We provide all information required to build the system: a solidworks assembly, part list, software and analysis code, along with technical specifications of the system in the Results section and Materials and methods sections *Optical system design, components and assembly* and *Patterned stimulation protocols*.

DeepLabCut. We have added additional details and discussion regarding the implementation of DeepLabCut to the Results section Sparse nociceptor stimulation triggers coordinated postural adjustments, and in the Discussion.

3. For widespread applicability and to determine the range, strengths, and weaknesses of this new tool to the pain field, the authors should extend their behavioral analyses. The reviewers preferred the authors to do as they mention in the discussion, which is to add an additional somatosensory line (perhaps a non-pain line) and see how their platform performs in comparison to Trpv1-ChR2. The less preferred option if the authors are not able to breed new somatosensory lines in reasonable time, is to try the Trpv1-ChR2 line in different contexts (inflammatory and/or neuropathic pain). In either case, at baseline or during chronic pain states in the Trpv1-ChR2 line, the authors should use an analgesic and show that their tool is modular and can detect decreases in pain-related signatures. The authors should take care to have N numbers closer to 10 animals per group, as the N of 4 in their studies is on the lower side.

This is an excellent suggestion. We have included an additional somatosensory Cre driver line, which took time to obtain and breed but substantially improves the manuscript. The somatosensory line we use is the Vglut1-Cre::ChR2-tdTomato line, which was originally characterized by Ru-Rong Ji and colleagues (Chamessian et al., J. Neurosci. 2019) in a relevant context. Here, we show that behavioral responses are specific to this stimulus. We therefore demonstrate the system is sensitive enough to detect behavioral responses associated with specific non-nociceptive and nociceptive input, and these responses can be readily discriminated.

Centrally acting analgesics (morphine, for example) may or may not suppress single-shot optogenetic stimuli; the interpretation of such an experiment does not address if pain is or is not experienced by such a stimulus, as they typically have other effects on arousal and locomotion that confound responses to mild inputs. A grooming or sleeping mouse, for example, does not respond to noxious stimuli as often as when it is at rest (Browne et al., Cell Reports 2017). We have previously shown that local lidocaine injection can block the optogenetic-evoked responses in these TRPV1-Cre::ChR2 mice (Browne et al., Cell Reports 2017). This blocks the input from the periphery. Further, the editor has previously developed a mouse pain scale and showed clearly that optogenetic stimulation causes pain-like responses (Abdus-Saboor et al., Cell Reports 2019). Therefore, experiments with analgesics would not contribute further to the existing understanding here. Computational classification of pain vs non-pain behaviors is an exciting area of research but is outside of the purpose of this paper. Our goal is to describe an approach for "remote touch" in freely behaving mice and provide some clear examples of its utility.

We have substantially extended our behavioral analysis of the TRPV1-Cre::ChR2 mice not to discriminate between pain vs non-pain, but based on the excellent suggestion to examine whole-body movement “…using a vector instead of a scalar”. We have extended the DeepLabCut analysis and reworked Figure 4, providing in depth computational analysis of the behavioral trajectories, demonstrating coordinated head orienting and body reposition. This further demonstrates the analysis and insights that can be achieved using this technical advance.

Regarding n numbers, all experiments used between 7 and 12 mice with one exception; the patterned stimulation experiment in Figure 2F used 6 mice, and 3 littermate controls and 4 off-target controls to confirm that these stimuli were indeed specific. These controls together show no responses, supporting what is already known with optogenetic stimulation of this mouse line. We have made the numbers of mice used clearer throughout the manuscript.